# Latitudinal Variation in the Toxicity and Sexual Compatibility of *Alexandrium catenella* Strains from Southern Chile

**DOI:** 10.3390/toxins13120900

**Published:** 2021-12-15

**Authors:** Camilo Rodríguez-Villegas, Patricio A. Díaz, Pilar Riobó, Araceli E. Rossignoli, Francisco Rodríguez, Patricia Loures, Ángela M. Baldrich, Daniel Varela, Alondra Sandoval-Sanhueza, Rosa I. Figueroa

**Affiliations:** 1Programa de Doctorado en Ciencias, Mención Conservación y Manejo de Recursos Naturales, Universidad de Los Lagos, Camino Chinquihue Km 6, Puerto Montt 5480000, Chile; ambaldrich@gmail.com; 2Centro i~Mar, Universidad de Los Lagos, Puerto Montt 5480000, Chile; dvarela@ulagos.cl (D.V.); alondrasandovalsanhueza@gmail.com (A.S.-S.); 3CeBiB (Centre for Biotechnology and Bioengineering), Universidad de los Lagos, Puerto Montt 5480000, Chile; 4Instituto de Investigaciones Marinas (IIM-CSIC), Eduardo Cabello 6, 36208 Vigo, Spain; pilarriobo@iim.csic.es; 5Centro de Investigacións Mariñas (CIMA), Pedras de Corón, s/n. Apartado 13, 36620 Vilanova de Arousa (Pontevedra), Spain; araceli.escudeiro.rossignoli@xunta.gal; 6Centro Oceanográfico de Vigo, Instituto Español de Oceanografía (IEO-CSIC), Subida a Radio Faro 50, 36390 Vigo, Spain; francisco.rodriguez@ieo.es (F.R.); patricia.loures@ieo.es (P.L.)

**Keywords:** dinoflagellate, saxitoxins, mating systems, resting cysts, PSP outbreak, Chilean Patagonia

## Abstract

The bloom-forming toxic dinoflagellate *Alexandrium catenella* was first detected in southern Chile (39.5–55° S) 50 years ago and is responsible for most of the area’s cases of paralytic shellfish poisoning (PSP). Given the complex life history of *A. catenella*, which includes benthic sexual cysts, in this study, we examined the potential link between latitude, toxicity, and sexual compatibility. Nine clones isolated from Chilean Patagonia were used in self- and out-crosses in all possible combinations (*n* = 45). The effect of latitude on toxicity, reproductive success indexes, and cyst production was also determined. Using the toxin profiles for all strains, consisting of C1, C2, GTX4, GTX1, GTX3, and NeoSTX, a latitudinal gradient was determined for their proportions (%) and content per cell (pg cell^−1^), with the more toxic strains occurring in the north (−40.6° S). Reproductive success also showed a latitudinal tendency and was lower in the north. None of the self-crosses yielded resting cysts. Rather, the production of resting cysts was highest in pairings of clones separated by distances of 1000–1650 km. Our results contribute to a better understanding of PSP outbreaks in the region and demonstrate the importance of resting cysts in fueling new toxic events. They also provide additional evidence that the introduction of strains from neighboring regions is a cause for concern.

## 1. Introduction

Toxic blooms of phytoplankton species are recurrent in Chilean coastal and offshore waters, occurring as a restricted local phenomenon or over a wide geographical area [1,2]. These harmful algal blooms (HABs) mainly consist of diatoms (e.g., *Pseudo-nitzschia* spp.) [3] and dinoflagellates (e.g., *Alexandrium catenella*, *Dinophysis acuta*, *D. acuminata*, *Protoceratium reticulatum*, and *Karenia selliformis*) [1,4,5,6]. The main causative organism of HABs is *A. catenella* and its negative repercussions on human health, ecosystems, fisheries, and aquaculture are well-documented [7,8]. The saxitoxins (STXs) produced by this dinoflagellate cause paralytic shellfish poisoning (PSPs), a severe to fatal illness induced in human consumers of contaminated seafood (wild or farmed) in which muscle paralysis is induced by the ability of STXs to block nerve impulses [1,7,9,10,11]. Consequently, outbreaks of *A. catenella* have serious economic impacts as they result in shellfish quarantines that adversely affect harvest, transport, and commercialization [12,13]. A reduction in toxin concentrations to safe levels after a bloom can take several months or even years [14].

The first *A. catenella* bloom in Bell Bay, in the southern-most austral region of Chile, Magallanes (53.9° S), was detected some 50 years ago [15]. Since then, the dinoflagellate has spread northward, with outbreaks in the Patagonian fjords system and the further coast including the Valdivian coast (~39.4° S) during an intense bloom that occurred in 2016 [16]. More recently, this species has been detected, albeit at relatively low cell abundances, in the Bio region (~36° S), an area never previously affected [17]. The broadening of the geographical distribution range of *A. catenella* by ~2500 km is surprising in terms of the magnitude of the blooms and their extreme toxicity [18,19]. Between 1996 and 2016, the toxicity of *A. catenella* blooms in Chile was characterized by a high interannual variability [12], with mussel toxicity reaching record levels of 143 × 10^3^ µg STX eq. 100 g^−1^ [20,21].

A prerequisite for the development of strategies aimed at predicting and possibly controlling *A. catenella* blooms is knowledge of the biological factors involved in bloom dynamics, especially those in the bloom termination phase when cells encyst and enter a resting stage [7,22]. The dinoflagellate mating system, which includes both self-fertilization (homothallism) and out-crossing (heterothallism), is an essential feature in the dinoflagellate encystment process. Homothallism involves the fusion of genetically identical gametes and therefore the crossing of clones, while in heterothallism, gametes from different genetic types, and thus two different clonal strains, fuse [23]. Either process results in sexual cysts (hereafter resting cysts) that are morphologically and functionally distinct from the cells of their vegetative counterparts [24,25,26]. Once the dormancy period is fulfilled and environmental conditions are appropriate, resting cysts germinate to become the inoculum of a new bloom event [7,26]. If the cysts are released into a new environment where they subsequently germinate, sexual recombination between this germinated population and the blooming genotypes already in that environment will give rise to a novel genetic pool. Therefore, an understanding of the origin of bloom events (i.e., the origins of the blooming strains) can allow forecasts of the toxicity of future bloom episodes.

*A. catenella* is a complex heterothallic species, as strains mate only with other compatible strains, with compatible pairs differing in their affinity and in their capacity for resting cyst production [27,28]. Mardones et al. [28] found a strong positive correlation between the genetic distance of parental strains and their reproductive compatibility. This correlation may be highly relevant for aquaculture management, since the movement of fish, shellfish stocks, fishing nets, and other aquaculture supplies can introduce new *A. catenella* strains from one region to another [29].

Despite the importance of *Alexandrium* species, there is little documentation of their biogeography, diversity, and ecology in the Southern Hemisphere [30]. This Special Issue of *Toxins*, therefore, provides an opportunity to fill this knowledge gap through studies of the differences in the toxicity of different *Alexandrium catenella* blooms and the role of the mating system in these differences. In this study, we analyzed the latitudinal patterns of strains isolated along the Chilean Patagonia fjord system.

## 2. Results

### 2.1. Molecular and Morphological Identification

According to their ITS rDNA sequences, the nine studied strains of *A. catenella* from southern Chile (Figure 1A,B) can be grouped in the same clade as other strains from different geographical origins, but were clearly differentiated from other related groups (i.e., *A. tamarense*, *A. australiense*, *A. pacificum*, and *A. mediterraneum* (Figure 1C). Optical and confocal microscopy showed cells displaying the thecal plate paratabulation typical of the species including the absence of the ventral pore (Figure 1D). Morphological changes during growth were also monitored (see vegetative division in Figure 1H,I) and confirmed the existence of sexuality and resting cyst formation. During these latter processes, morphologies only possible during sexuality were recorded (e.g., gamete fusion and zygote formation, Figure 1J) and sexual resting cysts were identified by calcofluor staining, based on the total lack of thecal plate paratabulation and the ovoid shape of the cells (Figure 1K).

### 2.2. PSP Toxins

HPLC analysis revealed detectable levels of six PSP toxins (hereafter PSTs) in all *A. catenella* strain cultures, with high intra-specific variability. Among the detected toxins were C1, C2, GTX4, GTX1, GTX3, and NeoSTX. GTX3 and C2 were the most abundant and NeoSTX the least abundant, detected in seven of the nine strains including in four in only trace amounts (<1%) (Figure 2A). The toxicity per cell (pg STX eq cell^−1^) and the proportions of the different toxins varied according to the geographical origin of the strain. For all strains, the median toxin concentration was 6.76 pg STX eq cell^−1^ (IQR = 10.80), with a maximum of 50.9 pg STX eq cell^−1^ (BM368) and a minimum of 2.59 pg STX eq cell^−1^ (MagI3) (Figure 2A). The linear model indicated an increase in the toxicity of *A. catenella* strains from south to north. This trend was not statistically significant at the 95% confidence level (observed *p* = 0.08). The adjusted model was more informative than the null model, based on the Akaike information criterion (AIC) (Figure 2B, Table 1).

A detailed analysis showed that strain toxicity increased from Magallanes (median of 2.76 pg STX eq cell^−1^; IQR = 0.17) to Aysén (8.89 pg STX eq cell^−1^; IQR = 5.61) and was highest in the Los Lagos region (median of 25.47 pg STX eq cell^−1^; IQR = 22.84) (Figure 3). These results indicated a northward latitudinal pattern in the toxicity of *A. catenella* strains within a broad geographical area in southern Chile.

### 2.3. Strains Sexual Compatibility

The results of self-cross and outcrossing experiments among the clonal strains of *Alexandrium catenella* are summarized in Table 2. There was no evidence of homothallic behavior in any strain because none of the self-crosses produced cysts. Complex heterothallism was assumed in *A. catenella*, given that the outcomes of crossing of compatible strains could not be explained by a simple +/− assignation. Ten out the 45 strain combinations produced resting cysts, resulting in a relatively low total compatibility index (TCI) of 22.2%. Resting cyst production was highly variable among the outcrosses and ranged between 1 and 98 cysts mL^−1^. These values were those of the compatible pairings Q119 × Ays12 and BM367 × Ays12, respectively (Table 2).

The reproductive compatibility indexes for the studied *A. catenella* strains showed latitudinal variation (Figure 4A). The low values of the compatibility index (CIs) in all strains except Ays12 (CIs = 0.67) and MagI3 (CIs = 0.56) derived from the low number of successful crosses possible between them. The average vigor (AVs) values, which are usually ≤2, indicated low resting cyst production in compatible crosses such as those of BM367 and Aysim57 (Figure 4A). The highest reproductive compatibility (RCs) value was obtained for strain Ays12 (RCs = 1.78), followed by MagI3 (RCs = 1.56) and BM368 (RCs = 0.78) present in the southern and northern ends of the study area, respectively (Figure 4A). In the linear model, the CIs, AVs, and RCs values tended to increase from north to south (CIs: *p* = 0.17, AVs: *p* = 0.16, and RCs: *p* = 0.13). This model, which was more informative than the null model, based on the AIC (Figure 5, Table 1), showed that the likelihood of resting cyst production by *A. catenella* tends to increase from north to south, albeit with a wide geographical variation.

Intra-regionally or between populations compatibility indexes (CIp, AVp, and RCp) results revealed a latitudinal pattern, with the highest RCp in the south (Mag × Mag, RCp = 0.93) and the lowest in the north (Los Lagos × Los Lagos, RCp = 0.40) (Figure 4B). However, according to the inter-regional results, the RCp was higher in strains from the southern regions (Aysén × Magallanes, RCp = 0.78) but similar for those from other regions (Los Lagos × Aysén, RCp = 0.56, and Los Lagos × Magallanes, RCp = 059) (Figure 4B).

When summarized in five geographical distance categories, resting cyst production revealed a latitudinal pattern (south–north) characterized by increasing densities of resting cysts (cysts L^−1^) arising from compatible pairings of strains separated by 0–100 km and 1000–1650 km (Figure 6A). In crosses of near to mid-distance (0–600 km) strains, the median resting cyst density was 0.0 cysts L^−1^, although cyst production ranged from ~1000 to 98,000 cysts L^−1^ (Figure 6A). Crosses that involved strains at distances of 601–1000 km and 1001–1650 km yielded a median of 1000 cysts L^−1^ (IQR = 7500) and 10,000 cysts L^−1^ (IQR = 29,500), respectively (Figure 6A). These results showed a geographical distance effect for *A. catenella* resting cyst production, in which larger distances between compatible pairings tended to produce higher abundances of resting cysts. All of the *A. catenella* resting cysts had a double cell wall, a red-brown body, and were oval-shaped (Figure 6B).

## 3. Discussion

The geographic spread of toxic dinoflagellates far from their native latitudes is a growing problem worldwide that is thought to be linked to anthropogenic factors including global warming [31,32]. This has been demonstrated in Puget Sound (Washington State, USA), where the increase in mussel toxicity has mostly been attributed to large-scale climate variability, which facilitates the spread of *A. catenella* [33,34].

This invasive behavior has caused the emergence of toxic syndromes such as PSP to latitudes never affected before. In Chile, *A. catenella* populations have followed a south to north expansion, and threats to aquaculture and human health have increased accordingly [17]. In addition to the role of higher sea temperatures in this expansion [33], other factors should be considered to better understand and accurately predict the latitudinal progression of this toxic dinoflagellate species [35,36].

Evolutionary adaptation studies conducted at the population level have shown the importance of sexual reproduction in the spread of invasive species, given that sexual recombination provides the genetic variability essential for adaptation and the establishment of new populations [37]. Once a species colonizes a new environment, clonal growth over the short-term is fast, resulting in the rapid expansion of established populations. However, clones eventually crash because they lack adaptability, such that a new sexual cycle is required for a sustained long-term response [38]. Therefore, in this study, we focused on the sexual compatibility of *A. catenella* clones isolated from different latitudes and the impact on toxicity. Our results revealed a clear latitudinal trend in both sexual compatibility and toxicity. The details and implications are discussed below.

### 3.1. Sexual Compatibility Rises from North to South

In the dinoflagellates, resting cyst production is a successful strategy for overcoming adverse environmental conditions, as a dormant stage is maintained until growth is environmentally supported [26,39]. Moreover, resting cysts are often a key factor in bloom recurrence as they provide the inoculum for new proliferative events. In the *A. catenella* life cycle, resting benthic cysts are derived from sexual processes (i.e., sexual induction/gamete differentiation, gamete recognition, and mating/zygote formation). Sexual induction appears to be species-specific and at least partly linked to the lack of certain nutrients (see revision by [40]), but the mechanism of gamete recognition (mating) and its regulation are thus far unknown. The mating behavior of *A. catenella* is complex, involving an unknown number of mating types and different degrees of affinity between compatible pairs. Several indexes have been proposed to evaluate mating affinity [41], summarized by the RC. The highest RC found in the studied *A. catenella* strains (RC = 1.78, in Ays 12) was lower than the RCs reported by Parker [42] in a study of Australian strains (Sydney Harbor, RC = 2.20) and similar to those reported by Mardones et al. [28] in their study of other Chilean strains (RC = 1.75, in Aysén and Los Lagos region). Although little is known about the sexual triggers represented by the RC values, genetic differences between parental strains seem to be a factor fostering sexuality in *A. catenella* [28] and other dinoflagellates [43]. This genetic difference could explain the high reproductive compatibility of Ays 12, as both traits turned out to be inversely correlated. Given that this strain comes from cyst germination, this genetic difference could be partially attributed to genetic recombination during meiosis [44]. On the other hand, the genetic difference with the other Aysén strains, all isolated from cysts, could be attributed to Ays 12 possibly coming from a cyst of a different cohort/bloom, or be a recently introduced strain from another locality. Indeed, evidence of successive cohorts has not been observed in the region [45]. Genetic differences between strains from the same locality, which segregate separately, were also observed previously in the same locality for the same gene sequence [30]. The genetic population structure of *A. catenella* from southern Chile showed a latitudinal gradient of genetic diversity, with higher diversity values in strains from the Aysén region and lower values from more northern strains [45], but our results showed a latitudinal trend of an increasing RC from north to south (Figure 5). This is not paradoxical, and can explain why the highest reproductive compatibility was found in Aysén (Ays 12), followed by Magallanes (MagI3) as well as why the predominant origin of the last intense PSP outbreaks also initiated in the Aysén region. In other words, it is similar to the focus of RC, and high genetic diversity converged in the Aysén region. This observation can also be supported by the higher genetic diversity found by Cruzat et al. [46] from in situ sampling analyses, and the higher *A. catenella* RC demonstrated by Mardones et al. [28] for this area.

### 3.2. Latitude, Toxicity, and RC: Relationship and Ecological Implications

PSP is endemic in the south austral region of Chile. Our results evidenced not only the increase in toxicity from south to north, but also the intra-specific and sympatric variation in toxicity among the studied strains of *A. catenella* including some strains isolated from the same PSP outbreak such as in 2016. A northward gradient in *A. catenella* cell toxicity was also found among *Alexandrium* populations from the northeastern coast of Canada and the United States (40–50° N) [47,48]. These studies determined a significant positive linear correlation between latitude and toxicity, with R^2^ values of 0.15–0.23, similar to the value obtained in this study for a latitude range of 40–55° S (R^2^ = 0.36, Figure 2B).

Our results are in line with those of other studies based on Chilean strains isolated from Canal Costa in the Aysén region in 1994 [49]. Strains isolated in 2004 and 2009 [30,50] showed similar intra-specific variation in their PSP toxin profiles, but in all cases, the highest toxicity was registered in strains isolated from Aysén to Quellón, which corresponds to the northern boundary of the geographic distribution of *A. catenella* in 2009. However, Aguilera-Belmonte et al. [50] detected GTX5 and STX, and more recently, Paredes-Mella et al. [17] obtained PST profiles of *A. catenella* strains from samples collected at 36° S (new northern limit of distribution). The latter study found large amounts of GTX5, which was not detected in this study.

The variation in toxin concentrations among *A. catenella* strains isolated from the same geographical area may reflect the dinoflagellate’s ecological persistence. For example, in the open ocean, strains with different PSTs profiles might be introduced from neighboring regions through active/passive spreading via aquaculture equipment (e.g., nets), the movements of shellfish from one location to another [29], resting cyst germination from unexplored areas (e.g., continental shelf [51]), or invasion, as suggested by Ichimi et al. [52]. This could explain the differences in the PST profiles of the sympatric strains BM367 and BM368 (isolated in 2016) as well as the similarity of the toxin profiles of strains BM367 and Ays 29 and of strains BM368 and Ays12. These patterns suggest a northward dispersion from Aysén to the Los Lagos region, but also the presence of diverse planktonic populations from the surrounding area with different PST profiles.

Unmeasured factors may also account for the toxin variation identified in this work. For instance, genetic diversity among strains with respect to *stxA4* gene copies [53], thought to be genetically fixed in each clonal strain [47,54], or nutrient limitation [55], defense responses to parasites/grazers [56,57], temperature/salinity [54,58], intracellular arginine/glutamate concentration [7], and the availability of light for toxin biosynthesis [59]. For this reason, further studies aimed at elucidating the potential role of these factors are needed to fully understand toxin biosynthesis by *A. catenella*. Our identification of latitudinal trends in toxicity and sexuality contribute to this goal by providing insights into the evolution of *A. catenella* blooms and their regional expansion. These results may in turn allow for the design of better and more reliable predictive models.

## 4. Conclusions

Our study demonstrated the existence of latitudinal trends in the toxicity and sexual affinity of the studied Chilean strains of *A. catenella*. Toxicity was shown to increase from south to north. In addition, intra-specific and sympatric variation in toxicity among strains, even those from the same bloom event, were determined. The identified increase in reproductive compatibility from north to south is in line with what is currently known about the genetic diversity of *A. catenella*. Our observations can contribute to more accurate predictions of *A. catenella* blooms, which may alleviate their economic impact in the southern austral region and elsewhere.

## 5. Materials and Methods

### 5.1. Strains Origin and Culture Conditions

The *A. catenella* strains used in this study were isolated from plankton samples or from the controlled germination of resting cysts carried out as described in [30]. The strains were collected at different sites covering a large geographical area in southern Chile (40°–55° S; ~1650 km) (See Figure 1B). Strains from Aysén (Ays12, Ays29, Aysim52, Aysim57) and Quellón (Q119) were isolated from resting cysts, and those from Magallanes (MagA7 and MagI3) and Bahía Mansa (BM367 and BM368) from plankton samples.

Stock cultures of each strain were cultivated in 125-mL Erlenmeyer flasks containing L1 medium without silicates and maintained at 15 °C ± 1 °C, with an irradiance of 90 µmol photons PAR m^−2^ s^−1^ and 12:12 h light–dark cycles [60]. The salinity of the L1 medium was that of Atlantic seawater, adjusted using sterile bi-distilled water to reach 31–32 psu.

### 5.2. DNA Extraction, PCR Amplification, and Sequencing

Exponentially growing vegetative cells from nine *A. catenella* strains were harvested by centrifugation (1.5 mL, 13,000 rpm for 2 min). The pellets were washed with sterile MQ water, centrifuged again, and stored at –20°C until further processing. DNA was extracted using the Chelex procedure described in [61]. The internal transcribed spacer (ITS1 and ITS2) and 5.8SrRNA gene regions were amplified in a PCR using the primer pair ITSF01/PERK-ITS-AS (5′-GAGGAAGGAGAAGTCGTAACAAGG-3′/5′-GCTTACTTATATGCTTAAATTCAG-3′ [62,63].

The 20-µL amplification reaction mixtures contained 10 µL of Horse Power Taq DNA polymerase master mix (Canvax Biontech, Córdoba, Spain), 0.5 µM of each primer, and 2 µL of the Chelex extracts. The DNA was amplified in an Eppendorf Mastercycler EP5345 under the following conditions: 5 min initial denaturing at 94 °C, followed by 30 cycles of 35 s denaturation at 94 °C, 35 s annealing at 55 °C, 1 min of elongation at 72 °C, and a final elongation step of 7 min at 72 °C. An 8-µL aliquot of each PCR was checked by agarose gel electrophoresis (1% TAE, 75 V) and Green Safe DNA gel stain (Canvax Biontech, Córdoba, Spain).

The PCR products were purified using ExoSAP–IT (USB, Cleveland, OH, USA), sequenced using the Big Dye Terminator v3.1 Reaction Cycle Sequencing Kit (Applied Biosystems, Foster City, CA, USA), and separated on an AB 3130 sequencer (Applied Biosystems) at the CACTI sequencing facilities (Universidade de Vigo, Spain). The complete ITS (ITS-1, ITS-2) and 5.8S rDNA sequences obtained in this study (and flanking regions of SSU and LSU rDNA) were deposited in the GenBank database (acc. nos. in Figure 2).

The sequences of the studied strains and those of related taxa from the genus *Alexandrium* obtained from Genbank were aligned using MEGA X software. The final alignment for the ITS phylogeny consisted of 529 positions. The phylogenetic model was selected using MEGA X. A Hasegawa–Kishino–Yano model [64] with a gamma-shaped parameter (γ = 1.28) was selected. The phylogenetic relationships were determined according to the maximum likelihood (ML) method using MEGA 6 and the Bayesian inference (BI) method with a general time-reversible model from Mr.Bayes v3.1 [65]. The two methods rendered very similar topologies. The phylogenetic tree was represented using the ML results, with bootstrap values from the ML method (*n* = 1000 replicates) and posterior probabilities from the BI method.

### 5.3. Morphological Identification

The samples were fixed with formalin (1.5%) and centrifuged at 5000× *g* for 10 min. The cell-containing pellet was resuspended in 2.5 mL of cold methanol and then stored for at least 12 h at 4 °C to facilitate pigment extraction. The cells were then washed in PBS (pH 7, Sigma-Aldrich, St. Louis, MO, USA) using the same centrifugation conditions, and the resulting pellet was resuspended in a staining solution consisting of 300 µL of a propidium iodide (Sigma-Aldrich, 60 µg·mL^−1^) and calcofluor white M2R (Sigma-Aldrich, 0.05 mg·mL^−1^) mixture for at least 2 h in darkness. The stained cells were then observed by optical fluorescence microscopy (LEICA MLA) and confocal microscopy (LEICA SP8) with excitation wavelengths of 488 and 405 and bright field (PMT trans) in the transmitted light. Imaging was performed at 63× magnification using the super-resolution mode LIGHTNING. Plate paratabulation was analyzed in strains Q119 and BM368, and sexuality in the crosses BM368x MagI3 and BM367xAys12.

### 5.4. Sexual Compatibility Study with Isolated Clonal Strains

The nine clonal *A. catenella* strains were self-crossed and out-crossed in all possible combinations (Table 2). Crosses were performed in duplicate in a final volume of 5 mL of L1-P medium (medium L1 with no phosphates added). The sexual induction protocol was that described in [43]. Thirty days after the crossing, the cultures were fixed in 1% paraformaldehyde. Resting cyst production was evaluated by counting the cysts present in 1 mL using a Sedgwick–Rafter chamber. Resting cysts formation served as the benchmark for reproductive compatibility and identification of the mating system [42]. The scoring criteria for resting cyst production (cysts L^−1^), based on [41], were as follows: (0) 0−3.0 × 10^2^; (1) >3.0 × 10^2^–2.0 × 10^3^; (2) >2.0 × 10^3^–1.0 × 10^4^; (3) >1.0 × 10^4^–1.0 × 10^5^; and (4) >1.0 × 10^5^. Self-crosses were considered homothallic, as the score for each one was 0. A total compatibility index (TCI) was calculated, defined as the percentage of compatible strains from the total pairings that produced resting cysts. Additionally, three indices of reproductive success [41] were calculated for each strain and for the intra-/inter-regional population groups from the data obtained from the crosses: the *compatibility index* (CI), defined as the number of compatible pairings resulting in a score ≥1 divided by the total number of possible crosses other than self-crosses; *average vigor* (AV), defined as the average of the cyst production scores (0–5) from all successful crosses involving a particular strain; and the *Reproductive Compatibility* (RC), calculated as the product of the CI and AV values. For the regional group populations, the RC indexes were calculated as the arithmetic means of the CIs and AVs of all intra- and inter-population crosses, as described in [41].

Finally, to identify a latitudinal pattern in *A. catenella* resting cyst production, cyst abundances were grouped into five categories according to the geographical distance: 0–100; 101–300; 301–600; 601–1000, and 1001–1650 km.

### 5.5. Toxin Extraction

Subsamples of 2 mL from each exponentially growing culture were fixed in acidic Lugol’s iodine solution and the cell concentration was then estimated using Sedgwick–Rafter chambers. Other subsamples of 40 mL of the same culture were filtered through GF/F glass microfiber filters 25-mm in diameter (Whatman, Maidstone, England) using a vacuum pump; the residual medium volume was estimated using glass graduated cylinders. Each cell-containing filter was placed in a 1.5 mL Eppendorf tube to which 750 µL of 0.05 M acetic acid was then added. The sample was then sonicated (1 min, 50 Watts), centrifuged at 17,968× *g* for 10 min, and the supernatant was transferred to a new 1.5-mL Eppendorf tube and processed as described above. The supernatants were combined (final volume 1500 µL) and kept at −20 °C until toxin analysis.

### 5.6. Analysis of PSP Toxins

PSP toxins were analyzed using high-performance liquid chromatography with post-column oxidation and fluorescence detection (HPLC-PCOX-FLD), according to a modification of the method in [66]. The LC column was a Waters X-Bridge Shield RP column (4.6 × 150 mm, 3.5 µm). Details of with the ultra-performance liquid chromatography equipment, post-column reaction system, FLD detector, reagents, mobile phases and the gradient conditions are described in Rodriguez et al. [67].

Data acquisition and data processing were performed using the Empower data system (Empower 3 Software, Copyright 2010 Waters Corporation ). Toxin concentrations were calculated based on the peak area and amount of each toxin as determined from the calibration curves, taking into account the toxicity equivalence factors proposed by the European Food Safety Authority EFSA [68]. The total concentration of PSTs in the samples was expressed as pg of STX equivalents per cell (pg STX eq cell^−1^). Different volumes were injected for each extract depending on the type of toxin analyzed. The limit of detection (LOD) was defined as the amount of toxin, which provided a response of three times the average area of the background noise peaks for the chromatogram basal line. LOD for GTX4, GTX1, GTX3, GTX2, and neoSTX was found to be 0.25, 0.2, 0.08, 0.2, and 0.12 ng on column, respectively. The limit of quantification (LOQ) was defined as the amount of toxin, which provided a response of 10 times the average height of the background noise for the chromatogram basal line. LOQ for GTX4, GTX1, GTX3, GTX2, and neoSTX was found to be 0.83, 0.66, 0.2, 0.66, and 0.4 ng on column, respectively. Standards for the PSP toxins gonyautoxin GTX-4, GTX-1, GTX-3, GTX-2, and neoSTX were acquired from the NRC Certified Reference Materials program (Halifax, NS, Canada). To verify the presence of N-sulfocarbamoyl-gonyautoxins 2 and 3 (C1,2 toxins), the samples were boiled with an equal volume of 0.4 M HCl for 15 min to hydrolyze the sulfonic group of the N-sulfocarbamoyl, yielding the corresponding carbamoyl toxins. The hydrolysis of C1 and C2 yields GTX2 and GTX3, respectively [69].

### 5.7. Data Analysis

A linear model (LM) was implemented to evaluate the influence of different latitudes on the toxin concentration (pg STX eq cell^−1^) and reproductive success indexes (CIs, AVs, and RCs) of *A. catenella* in the study area. Normality was checked by the Shapiro–Wilk test, homoscedasticity by Levene’s test, and collinearity by the variance inflation test (VIF) using a threshold VIF of ≤4 [70]. Data overdispersion was assessed by comparing observed and expected dispersion values [71,72]. Explanatory variable effects were tested using a sequential test -type I ANOVA [73]. Comparisons among the models (adjusted and null) were made based on the Akaike information criterion (AIC) [74]. Following the recommendations of the American Statistical Association [75] and an increasing number of scientists worldwide [76], we avoided a dichotomic use of *p*-values, which are provided as calculated.

## Figures and Tables

**Figure 1 toxins-13-00900-f001:**
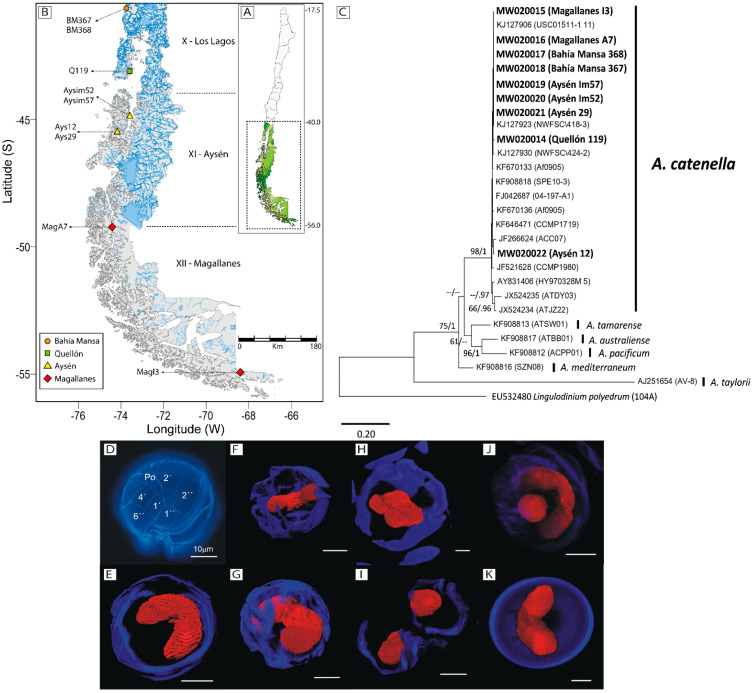
(**A**) Map of Chile, with the inset showing southern Patagonia. (**B**) The specific location of the *A. catenella* strains isolated from Los Lagos, Aysén, and Magallanes. (**C**) Phylogenetic relationships among *Alexandrium catenella* strains based on the ITS1 and ITS2 sequences obtained in this study and from GenBank. *Alexandrium taylorii* and *Lingulodinium polyedrum* sequences were used as outgroups. The phylogenetic tree was constructed using the maximum-likelihood method. Numbers at the branches indicate the percentage of bootstrap support (*n* = 1000) and posterior probabilities based on Bayesian inference as a search criterion. Bootstrap values <60% and probabilities <0.6 are denoted by hyphens. Names in bold represent isolates sequenced for this study. (**D**) Optical and (**E**–**K**) confocal) microscopy of *A. catenella* using specific fluorochromes for DNA (propidium iodide, red nucleus) and thecal (calcofluor, in blue) staining. (**D**) Vegetative cell in apical ventral view, showing the thecal plate paratabulation typical of *A. catenella*, (**E**–**G**) the typical U-shaped nucleus from different angles in relation to the external morphology of the cells, and (**H**,**I**) mitotic division in early (**H**) and late (**I**) stages. Note that during mitosis (desmoschisis type), the parental theca is shared between the two sister cells. Divided cells completely separate before the thecal plates are fully replaced (**I**). (**J**) Putative mobile zygote (planozygote) containing two nuclei, (**K**) resting cyst. Scale bars = 10 µm.

**Figure 2 toxins-13-00900-f002:**
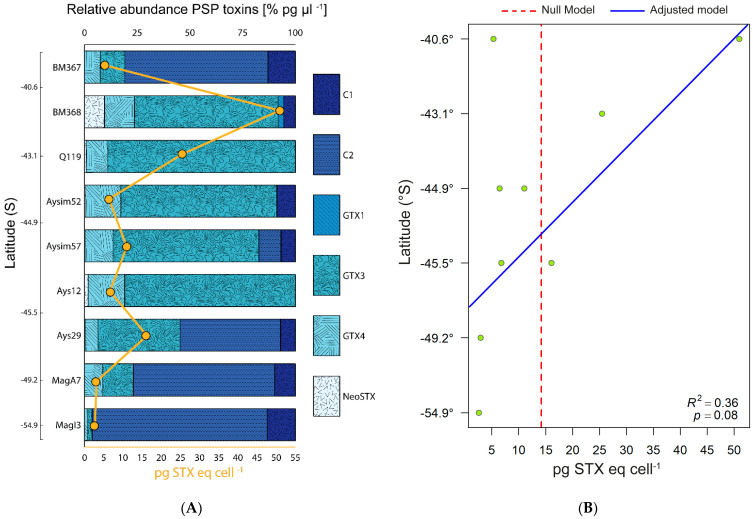
(**A**) Specific PSTs profiles of *Alexandrium catenella* strains from southern Chile. BM; Bahía Mansa. Q; Quellón. Ays; Aysén. Mag; Magallanes. All strains are ordered in the plot using latitudinal criteria from north (top) to south (bottom). (**B**) Linear models of *A. catenella* PSP toxins (pg STX eq cell^−1^): the null model (arithmetic mean) and the adjusted model.

**Figure 3 toxins-13-00900-f003:**
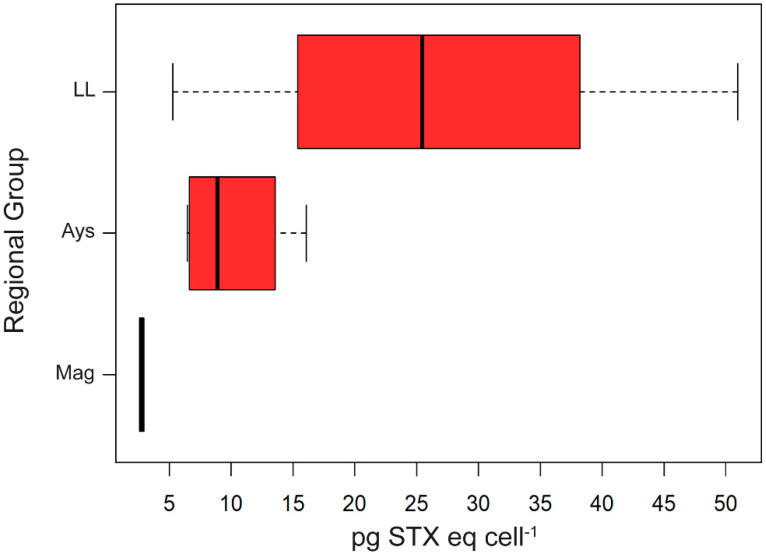
Boxplot of regional PST concentrations of *Alexandrium catenella* strains from southern Chile. LL; Los Lagos. Ays; Aysén. Mag; Magallanes. The boxplots show the range (whiskers), median (bold line), and interquartile range (box height). The strains are ordered in the plot using latitudinal criteria, from north (top) to south (bottom).

**Figure 4 toxins-13-00900-f004:**
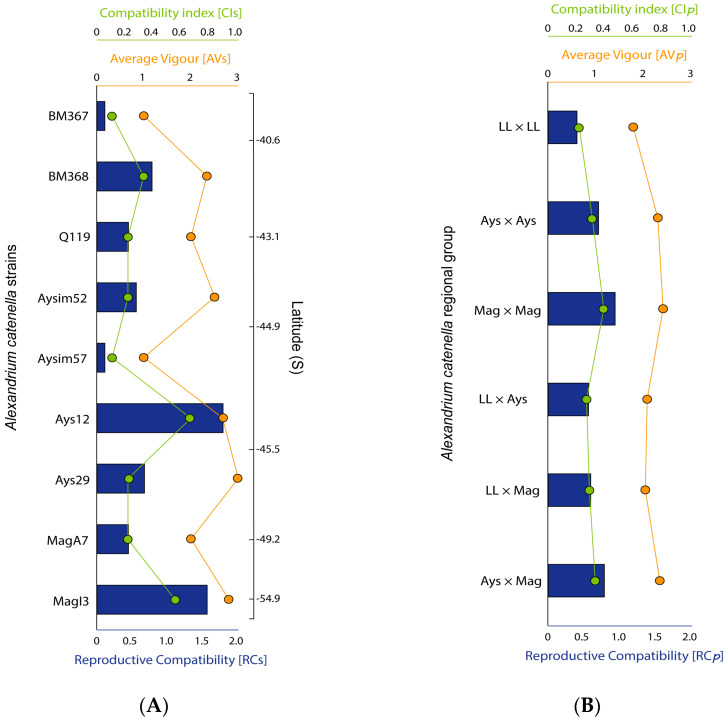
(**A**) Reproductive compatibility of each *Alexandrium catenella* strain based on the CIs, AVs, and RCs. The strains in the plot are shown in latitudinal order, from north (top) to south (bottom). BM; Bahía Mansa. Q; Quellón. Ays; Aysén. Mag; Magallanes; (**B**) Inter- and intra-population CIp, AVp, and RCp calculated as the arithmetic means of the CI and AV values of each inter- and intra-regional group population. LL; Los Lagos. Ays; Aysén. Mag; Magallanes.

**Figure 5 toxins-13-00900-f005:**
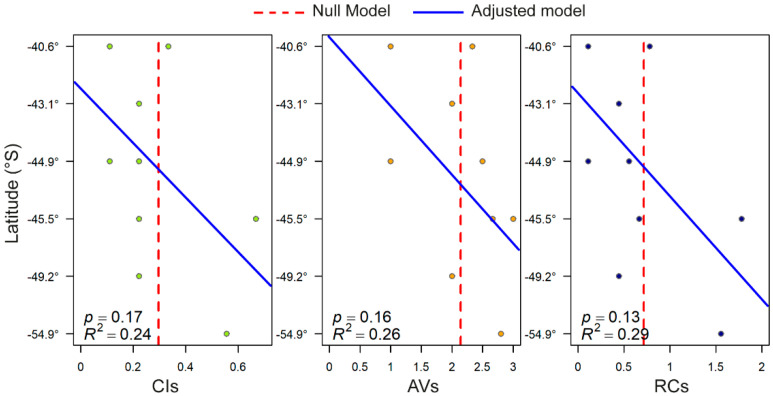
Linear models of the reproductive compatibility indexes (CIs, AVs, RCs) of *Alexandrium catenella* strains from southern Chile. The strains are organized in the plot according to their latitude, from north (top) to south (bottom). The null model represents the arithmetic mean of each reproductive index.

**Figure 6 toxins-13-00900-f006:**
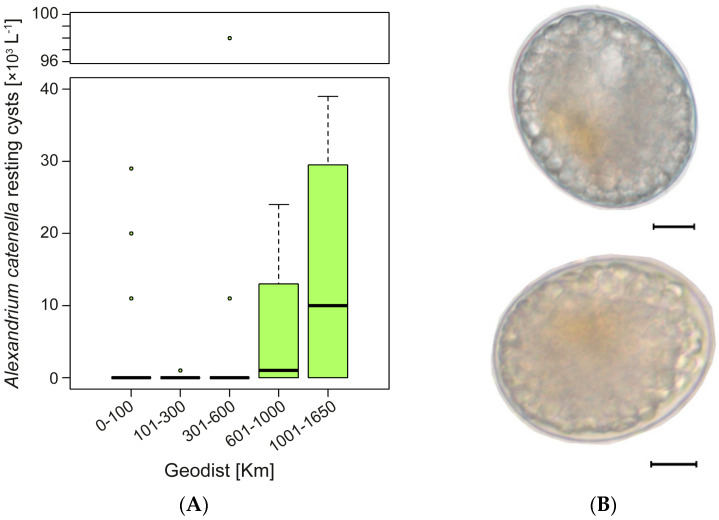
(**A**) Boxplot of the abundance of *Alexandrium catenella* resting cysts produced in compatible pairings from strains isolated at different geographical distances. The boxplots show the range (whiskers), median (bold line), and interquartile range (box height). (**B**) Optical microscopy of *A. catenella* resting cysts obtained in intercross experiments: top: Ays29 × MagI3; bottom: BM368 × MagI3. Scale bars = 10 µm.

**Table 1 toxins-13-00900-t001:** Statistical significance of the explanatory variables, determined using a type I linear models (ANOVA), for the latitudinal variability of *A. catenella* toxins (pg STX eq cell^−1^), compatibility index (CI), average vigor (AV), and reproductive compatibility (RC). The adjusted and null models (based on the arithmetic mean) for each variable were compared based on the AIC. * Denotes the most informative model.

Linear Model	Df	Sum Sq	Mean Sq	F Value	Pr (>F)	AIC
Toxins	1	715.5	715.50	4.016	0.083	75.822 *
Residuals	7	1233.3	176.19			
Null						575.99
CIs	1	0.071	0.071	2.219	0.179	1.660 *
Residuals	7	0.224	0.032			
Null						658.15
AVs	1	1.109	1.109	2.465	0.160	22.088 *
Residuals	7	3.148	0.449			
Null						608.24
RCs	1	0.801	0.801	2.890	0.132	17.738 *
Residuals	7	1.941	0.277			
Null						620.72

**Table 2 toxins-13-00900-t002:** Production of resting cysts by all possible crosses of the tested strains of *Alexandrium catenella*. The strains are ordered using latitudinal criteria, from the north (top and left) to south (bottom and right). The scoring criteria for resting cysts production are shown in bold and the raw data of resting cyst production (cysts mL^−1^) in parentheses (see Material and Methods). BM; Bahía Mansa. Q; Quellón. Ays; Aysén. Mag; Magallanes.

Regional Group	Strain	BM367	BM368	Q119	Aysim52	Aysim57	Ays12	Ays29	MagA7	MagI3
Los Lagos	BM367	0 (0)	0 (0)	0 (0)	0 (0)	0 (0)	3 (98)	0 (0)	0 (0)	0 (0)
	BM368		0 (0)	0 (0)	0 (0)	0 (0)	3 (11)	0 (0)	1 (2)	3 (25)
	Q119			0 (0)	0 (0)	0 (0)	1 (1)	0 (0)	0 (0)	3 (39)
Aysén	Aysim52				0 (0)	0 (0)	3 (20)	0 (0)	0 (0)	2 (10)
	Aysim57					0 (0)	3 (11)	0 (0)	0 (0)	0 (0)
	Ays12						0 (0)	3 (29)	0 (0)	0 (0)
	Ays29							0 (0)	0 (0)	3 (34)
Magallanes	MagA7								0 (0)	3 (24)
	MagI3									0 (0)

## Data Availability

Not applicable.

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
