# Peer review of "Latitudinal Variation in the Toxicity and Sexual Compatibility of Alexandrium catenella Strains from Southern Chile"

_toxins, 2021, doi:10.3390/toxins13120900_

Round 1

Reviewer 1 Report

Dear editor and authors,

The manuscript entitled "Latitudinal variation in the toxicity and sexual compatibility of Alexandrium catenella strains from southern Chile" (1469981) investigated the potential link between latitude, toxicity, and sexual compatibility of 9 Alexandrium catenella strains isolated from Chilean Patagonia. This study falls within the journal's scope. From my point of view, this work provides some interesting results, but the interpretation and conclusion about the link between latitude, toxicity and sexual compatibility is unconvincing, therefore, I have some concerns about this study detailed below.

Major concerns:

The manuscript needs a thorough revision to deal with a number of significant issues I identified, which affect all sections in the manuscript:

  1. While the manuscript is overall well-structured and concise, there are many grammatical and phrasing issues, sometimes linked to incorrect usage of tense and clauses. There are long sentences with inaccurate clauses which should be avoided, such as at line 107-110. This issue extends beyond what might be expected for a reviewer to correct, so I suggest additional efforts/interventions are required to improve the English written presentation.
  2. One suggestion is to put the figure 7 as the first figure at the beginning of the manuscript to introduce the sampling location of the strains, as the original figure 1 to figure 6 all have related latitude information which need to be clearly presented at the beginning. For example, in figure 1a, it is better to compare the phylogeny of strains with their geographic locations. The authors need to explain why Aysen 12 is not clustered with other Aysen strains, even it is located together with Aysen 29? Another suggestion is to add, for example, the current information or major fish migration event along the coast of southern Patagonia, which may provide more evidence in the discussion about the evolution and diversity of strains along distance.
  3. The authors mentioned “the toxin profiles were similar for all strains” in the abstract at line 11, but in figure 2a, the toxin profiles of the 9 strains are so diverse even between closely located strains (for example, BM367 and BM368, Asy12 and Asy29), please provide inferred information about the reason what makes this difference? Moreover, in figure 2a and 3, the authors concluded that the toxicity increased from south to north (even not significant P=0.08), which is probably mainly due to the high toxicity of the strain BM368, but the neighboring strain BM367 is still as lowly toxic as other strains. This is the reason why there is only a trend between toxins and latitude, but still not significant. Therefore, these results could not support the conclusion solidly, and the authors may need to provide more evidence.
  4. In table 2, the highest RC was detected for strain Ays12, together with the phylogenetic relationships in figure 1a, what makes the Ays12 different? And at line 234-240, the authors raised the discussion, but not clearly reached the point. Please provide more logic explanation. Moreover, in table 2, the authors provided both scoring criteria (bold) and raw data (in parentheses), to me, these two parts are generally the same information which is redundant and makes the table difficult to read.
  5. In figure 4 and 5, the similar problem as mentioned before, the trend of RC, AC, and CI with latitude is not as clear as the authors concluded, which was again could be seen in the not significant linear models (Figure 5). 

Reviewer 2 Report

Dear authors,

The present manuscript entitled "Latitudinal variation in the toxicity and sexual compatibility of Alexandrium catenella strains from southern Chile" examines the examined the potential link between latitude, toxicity, and sexual compatibility of several strains of A. catenella from Southern Chile.
The work is properly designed and the methodology used is adequate and well described. The literature survey and discussion is well documented.

This is an interesting and well-writing study and I think it makes a relevant scientific contribution to this research field and fits well with the aims and scope of this special issue of Toxins.

I wish especially to congratulate the authors for their great work and excellent presentation.

I support the publication of this study in present form.

Reviewer 3 Report

This article provides the scientific community with very interesting results to better understand the occurence of the blooms of A.catenella in Chile. You will find my remarks below.

  • line 51 : the concentration of PSP toxins is in mussel ?
  • line 86 : missing a space between "and" "A.mediterraneum"
  • line 117: figure 2.(a) it is not clear if you performed replicates for toxins analysis. I think you performed a single analysis of toxins content for each strains. In this case i think you must be less affirmative.
  • line 160 : "tend" to increased (i suggest you)
  • line 203 : add references (suggestion but not the only one:  Genovesi, B., Mouillot, D., Laugier, T., Fiandrino, A., Laabir, M., Vaquer, A., Grzebyk, D., 2013. Influences of sedimentation and hydrodynamics on the spatial distribution of Alexandrium catenella/tamarense resting cysts in a shellfish farming lagoon impacted by toxic blooms. Harmful Algae 25(0), 15-25.
  • lines 215-218 : add references 
  • M & M 5.6 Analysis of PSP toxins : can you indicate LD and LQ of the method
  • reference 32: Moore, S.K., Trainer, V.L., Mantua, N.J. et al. Impacts of climate variability and future climate change on harmful algal blooms and human health. Environ Health 7, S4 (2008). https://doi.org/10.1186/1476-069X-7-S2-S4

Round 2

Reviewer 1 Report

I am glad to see the efforts that the authors put to improve the manuscript, even there is still questions can not be clearly answered with the present data, it is an interesting study and could be published as the revised form.

Author Response

Thanks for your encouraging comments!